# Exploring the Clustering Property and Network Structure of a Large-Scale Basin's Precipitation Network: A Complex Network Approach

**Yiran Xu [1] , Fan Lu [1],\*, Kui Zhu [2],\*, Xinyi Song [1] and Yanyu Dai [1]**

[1]    State Key Laboratory of Simulation and Regulation of Water Cycle in River Basin, China Institute of Water Resources and Hydropower Research, Beijing 100038, China; hsuijan@163.com (Y.X.); ts14010083@cumt.edu.cn (X.S.); daiyanyu_dyy@163.com (Y.D.)

[2]    School of Resources and Earth Science, China University of Mining and Technology, Xuzhou 221116, China

\*    Correspondence: lufan@iwhr.com (F.L.); zkxjzkxj@163.com (K.Z.);
     Tel.: +86-10-687-816-56 (F.L.); Fax: +86-10-684-833-67 (F.L.)

**Abstract:** Understanding of the spatial connections in rainfall is a challenging and essential groundwork for reliable modeling of catchment processes. Recent developments in network theory offer new avenues to understand of the spatial variability of rainfall. The Yellow River Basin (YRB) in China is spatially extensive, with pronounced environmental gradients driven primarily by precipitation and air temperature on broad scales. Therefore, it is an ideal region to examine the availability of network theory. The concepts of clustering coefficient, degree distribution and small-world network are employed to investigate the spatial connections and architecture of precipitation networks in the YRB. The results show that (1) the choice of methods has little effect on the precipitation networks, but correlation thresholds significantly affected vertex degree and clustering coefficient values of precipitation networks; (2) the spatial distribution of the clustering coefficient appears to be high–low–high from southeast to northwest and the vertex degree is the opposite; (3) the precipitation network has small-world properties in the appropriate threshold range. The findings of this paper could help researchers to understand the spatial rainfall connections of the YRB and, therefore, become a foundation for developing a hydrological model in further studies.

**Keywords:** complex networks theory; precipitation; clustering coefficient; small-world network; topology

## 1. Introduction

The hydrologic cycle is a complex atmospheric and hydrological process. Its temporal and spatial changes are characterized by strong nonlinear characteristics, deeply influenced by many factors such as human activity, topography and geomorphology [1]. Precipitation, as an important part of the water recycling system, forms a key input in numerous climatic and hydrological studies, including catchment hydrological modeling and drought prediction. However, it has always been a great challenge to understand the spatial and temporal variability of rainfall completely, as affected by multiple factors such as climate, topography and land use. Some methods based on mathematical statistics are currently employed, which focus on the correlation, stationarity, periodicity, mutation and trends of hydrological time series, such as, Mann–Kendall trend test [2,3], and Wavelet analysis [4–6]. These methods focus on data itself, and pay less attention to the structural characteristics of stations. Meanwhile, complex network theory provides a new perspective to study the spatial variation of precipitation [7,8].

Many discoveries of complex networks up to now, such as basic models of network topology, propagation mechanisms of complex networks, and the synchronization behavior of complex dynamical

networks, make complex network theory widely used in all fields, including in large power networks, transportation networks, social networks and spreading networks [9–14]. However, the application of complex networks in the hydrological field is still in its primary stage and mainly focuses on the following three aspects. (1) The evolution of extreme events, such as heatwaves or rainfall. The event synchronization method (ES) is employed to quantify the synchronicity of extreme events. Network edges are placed between two nodes if the corresponding synchronization values are significant. Then, the indicators in complex networks, such as degree, clustering coefficient, closeness centrality and betweenness centrality, are adopted to analyze the spatial or spreading characteristics of extreme events [15–19]. For example, Boers [20] revealed the global coupling pattern of extreme rainfall events by applying a complex network methodology to high-resolution satellite data and introducing a technique that corrects for multiple-comparison bias in functional networks. (2) The detection of time-series variability, including precipitation or temperature series. The coarse graining process is employed to convert the data series into character sequences. A string consists of several characters represent nodes, and network edges are placed between two nodes according to the time sequence. Then, clustering coefficient, average path length, and the concept of a scale-free or small-world network is used to reveal climate change [21]. For example, Liu et al. [22] used the coarse graining process to convert the data series of daily mean temperature and daily precipitation from 1961 to 2011 into symbol sequences and created climate fluctuation networks for understanding the complexity of climate change. (3) Spatial connections of rainfall or runoff. For the rainfall spatial connections, linear correlation coefficient (Spearman or Pearson) is used to define edges between precipitation stations (nodes). Most studies focused on the spatial connections, temporal scale or network architecture [23–26]. For the connections in streamflow dynamics, except for linear correlation coefficient [27,28], the horizontal visibility approach is employed to construct the runoff network to exploit the duality between time series and networks, to investigate the dynamics of river flows, and to optimize hydrometric monitoring system design [29,30].

As mentioned above, the linear correlation coefficient is a common method to define edges. It is important to know how the selection of the correlation methods affects the network and which one is more suitable for the network construction. Halverson and Fleming [27] explored the impacts of using Spearman rank correlation in place of Pearson linear correlation when they constructed a streamflow network in British Columbia, Canada. They reported that Spearman correlation tends to increase the number of edges between stations but does not change the global network structure. Unfortunately, the conclusion was given in a few words without more details. There are still few studies on the detailed comparison between networks using different correlation methods. Correlation threshold (CT) is also crucial to network construction: what is the double influence of the correlation threshold and the correlation method on the number of edges, clustering coefficient values and network structure? In addition, it is also worth exploring what the explanation of spatial distribution of clustering coefficient is and inferring the architecture of a precipitation network. Therefore, extensive studies on the spatial connection of rainfall using complex networks still need to continue in many areas, especially some with complicated topography and variable climate, like in the YRB. This paper might help researchers to understand the spatial rainfall connections of the YRB and become a foundation for developing a hydrological model in our further studies. This provides the motivation for this study. Therefore, in the present study, we apply the concept of complex networks to analyze the spatial connection of rainfall stations using the clustering coefficient and test the influence of correlation methods and correlation thresholds on the precipitation network. We also study the characteristics of assertive mixing, including the relationship between vertex degree and clustering coefficient, and the relationship between stations with different vertex degrees. We then assess the small-world network characteristics. The rest of this paper is organized as follows. Section 2 presents a brief description of the small-world network and the procedure for calculation of degree, degree distribution, clustering coefficient and the average path length used in the present study. Details of the study area and rainfall data are presented in Section 3. Section 4 presents the details of network

construction, the impacts of the correlation methods and correlation thresholds on clustering coefficient values, and the network architecture. Section 5 gives the conclusions of this study.

## 2. Network Methodology

The study of complex networks has received increasing interest in recent years. As for various networks, we need an important tool, which is mathematically called a graph, to describe their structural characteristics. Representing entities and relationships between entities with vertices and edges respectively, a network can be considered as a set of vertices and edges in graph theory.

The simplest form of network consists of a few identical vertices connected by identical edges. However, a network may be highly complex in many ways. For instance, a network (1) may have millions of more than one type of vertices and/or edges; (2) may contain edges that have different weights and that can be directed; (3) may have various forms of edges, such as multi-edges, self-edges and hyperedges; and (4) may generate or lose vertices and/or edges over time. Some additional details about this can be found in B. Sivakumar [28].

Networks in hydrology may be less complex, so the existing methods and measures are competent enough to analyze the characteristics of networks in hydrology. Degree, degree distribution, local/global clustering coefficient, average path length and small-world networks are some of the basic and important concepts. They are described next so that readers can understand the theories of this article.

### 2.1. Degree and Degree Distribution

We assume that a network is defined by a set of $V = 1, \ldots, N$ vertices and a set of E edges$\{i, j\}$. Edges of the network are undirected and no self-edges $\{i, i\}$ are allowed; that is to say, there can be at most one edge between two vertices. The adjacency matrix, A:

$$A_{ij} = \begin{cases} 0, & \text{if}\{i, j\} \notin E \\ 1, & \text{if}\{i, j\} \in E \end{cases} \qquad (1)$$

A takes into account whether an edge is active or not between vertices i and j. Since the network is considered undirected and no self-edges are allowed, A is symmetric and $A_{ii} = 0$.

The degree of a vertex is its most basic structural property, the number of its adjacent edges. Intuitively, the greater the degree of a vertex means the more important it is in a certain sense. For instance, a vertex has four edges and its degree is $k = 4$. The degree distribution, p(k), describes the distribution of the vertex degree of the network, which gives the probability of a randomly selected vertex having exactly k edges. As to a random network, whose edges are placed randomly, the degree distribution is a Poisson distribution. The majority of vertices have approximately the same degree and are close to the average degree. In recent years, a large number of studies have shown that the degree distribution of many real networks is obviously different from the Poisson distribution [31,32]. Some can be described better by a power-law distribution ($p(k) \propto k - r$). These networks can be called scale-free networks. Similarly, degree distribution can also be exponential distribution ($p(k) \propto e - k / k$) [33]. In particularly, the power-law distribution corresponds to a straight line in the logarithmic coordinate system, while the exponential distribution corresponds to a straight line in the semi-logarithmic coordinate system. Therefore, they can be easily recognized by logarithmic coordinates and semi-logarithmic coordinates, respectively.

### 2.2. Clustering Coefficient

The local clustering coefficient of a network is basically a measure of local density. The procedure for calculation of the local clustering coefficient is as follows. We assume that vertex i in the network is connected to $k_i$ other vertices via $k_i$ edges, and the $k_i$ vertices can be called neighbors of vertex i. Obviously, there would be $k_i(k_i - 1) / 2$ edges between neighbors. The clustering coefficient of vertex i

is then given by the ratio between the number $E_i$ of edges that actually exist between these $k_i$ vertices and the total number $k_i(k_i - 1) / 2$,

$$C_i = \frac{2E_i}{k_i(k_i - 1)} \qquad (2)$$

The global clustering coefficient C is the average of the local clustering coefficients of all the individual vertices ($0 \leq C \leq 1$). The global clustering coefficient of a completely ordered network equals 1.0, while a global clustering coefficient of 0 indicates a network without any edges. In addition, the global clustering coefficient of a random graph is $C = p$ (where p is the probability of two vertices being connected), while for a completely random network, with N vertices, its global clustering coefficient is $C = N^{-1}$.

*2.3. The Average Path Length*

The distance from vertex i to vertex j, $d_{ij}$, is the minimum number of edges that have to be crossed from vertex i to vertex j. The maximum distance between any two vertices can be called the diameter of the network (D). The average path length is the average distance between any two vertices:

$$L = \frac{2}{N(N+1)} \sum_{i \geq j} d_{ij} \qquad (3)$$

where N is the number of vertices. For convenient mathematical treatment, Equation (3) contains the distance from vertex i to itself (of course, $d_{ii} = 0$). The error can be ignored when N is very large.

*2.4. Small-World Network*

Before introducing small-world network model, we introduce a fundamental network model, the random network model, which is arguably the most well-developed class of network graph models, mathematically speaking [34]. The classical theory of random graph models, as established in a series of seminal papers by Erdős and Rényi [35–37], rests upon a simple model that places an equal probability on all graphs of a given order and size. Random networks have a small clustering coefficient and a small average path length. Although they are useful idealizations, they cannot embody some important features of real networks and they are not often observed in real-world phenomena. In fact, many real-world networks have a small average shortest path length, but also a clustering coefficient significantly higher than expected by random chance, such as electric power grids, metabolite processing networks, networks of brain neurons and social influence networks.

For this reason, D. Watts and S. Strogatz [38] introduced an interesting model called WS small-world network model, which is explicitly designed to mimic certain observed "real-world" properties. Small-world networks tend to contain sub-networks, which have connections between almost any two vertices within them. This follows from the defining property of a high clustering coefficient. Secondly, most pairs of vertices will be connected by at least one short path, which means the average path length is small. Network small-worldness is quantified by a small coefficient, σ, calculated by comparing clustering and path length of a given network to an equivalent random network with the same degree on average.

$$\sigma = \frac{C/C_r}{L/L_r} \qquad (4)$$

where $C_r$ and $L_r$ are the clustering coefficient and average path length of the equivalent random network respectively. If $\sigma > 1$ ($C \gg C_r$ and $L \approx L_r$), the network has small-worldness.

## 3. Study Area and Data

In this study, the Yellow River Basin (YRB) was selected as a case study region to explore the effectiveness of the complex network theory for identifying spatial connections in precipitation.

The Yellow River is the sixth longest river in the world at the estimated length of 5,464 km. Its total drainage area is about 795,000 km². There are significant differences in the spatial and seasonal distribution, inter-annual variations of precipitation in the YRB, which lies in that (1) the YRB is in the north of the East Asian monsoon region, and parts of the region are also affected by the plateau monsoon. (2) The terrain of the YRB is extremely complex, with a great disparity in height between the east and west parts. (3) The underlying surface conditions are complex. According to Chang's study [39], the mean annual rainfall across the whole basin is about 446.27 mm (1960–2010). The precipitation decreases from southeast to northwest and increases from upstream to downstream.

In the present study, monthly rainfall data from a network of 379 stations across the YRB are considered for analysis. Considering the vast area of the YRB and the influence of the YRB's boundaries, it is necessary to have a large number of rainfall stations in and around the YRB to analyze spatial precipitation variability. Limited to the quality of the precipitation data, including its integrity and reliability, we selected 379 stations from hundreds of rainfall stations in the YRB, whose monthly precipitation time series were observed for a period of 56 years (1956–2012). The data is obtained from the China meteorological data service center (CMDC) [40]. The 379 selected stations and their observed precipitation data exhibit considerable variations in their characteristics: (1) station elevation ranges from 0 to 6065 meters (see Figure 1). From west to east, the YRB could be roughly divided into three parts. The highest part is located in the northeast of the Qinghai Tibet Plateau, with an average elevation of over 4000 meters. The second part lies in the Loess Plateau, and the altitude ranges from 1000 to 2000 meters. The third part is below 100 meters, located in the North China Plain. (2) The average annual precipitation ranges 200–650 mm, while it is more than 650 mm downstream and less than 150 mm in the northwest.

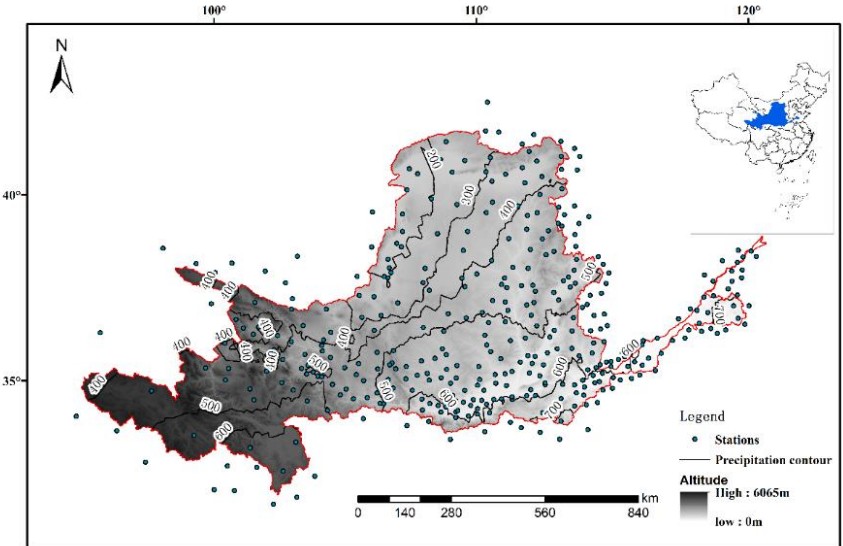

**Figure 1.** Locations of stations and contour of annual average rainfall in the YRB (data from CMDC [40]).

## 4. Analysis and Results

### 4.1. Network Construction

How to construct a network? The most fundamental problem is edge definition. For example, in a traffic network, roads or railways define the edge between cities. However, in the case of hydrology application, there might not be a straightforward binary relationship between vertices, meaning it becomes necessary to consider empirical relationships. Malik [15] employed event synchronization (ES) as a nonlinear correlation to measure the strength of synchronization of rain events between two different grid points and analyzed summer monsoon rainfall over the Indian peninsula. ES is applicable to the analysis of spatial connections and propagation of extreme events. A more common method

to define edges between precipitation observed at different stations is through a cross correlation analysis. We assign an edge between a pair of stations when their correlation coefficient, r, exceeds some threshold, rt. In the present study, we employed two methods, the Pearson and Spearman correlation methods, to define edges. The Pearson correlation evaluates the linear relationship between two continuous variables. The Spearman correlation evaluates the monotonic relationship between two continuous or ordinal variables. In a monotonic relationship, the variables tend to change together, but not necessarily at a constant rate. The Spearman correlation coefficient is based on the ranked values for each variable rather than the raw data. If edges are defined by a threshold correlation coefficient, then we naturally consider which threshold to choose and different thresholds' impacts. We consider nine different values of correlation threshold (CT): 0.5, 0.55, 0.6, 0.65, 0.7, 0.75, 0.8, 0.85, 0.9. Therefore, with two correlation methods (P-network and S-network) and nine correlation thresholds, there are a total of 18 precipitation networks to conduct the sensitivity test.

### 4.2. Descriptive Analysis of Network Graph Characteristics

The clustering coefficient (CC) is calculated for each of the above 379 rainfall stations in the YRB, following the procedure described in Section 2.2. In general, the correlation threshold may sometimes significantly influence the clustering coefficient, and there is an inverse relationship between them. Therefore, we choose nine threshold values to assess their influence and interpret the results. We also attempt to connect the spatial distribution of the clustering coefficient with annual average precipitation and vertex degree. It is noted that clustering coefficient, in this section, means local clustering coefficient.

Figure 2, for instance, presents the clustering coefficient values of a P-network for nine threshold values. It must be said that the clustering coefficient value of NA is completely different from 0, which means that a station has no nearest neighbors; that is to say, the $k_i$ in Equation (2) is 0. However, a clustering coefficient value of 0 indicates that a station has several neighbors but there are no edges between them.

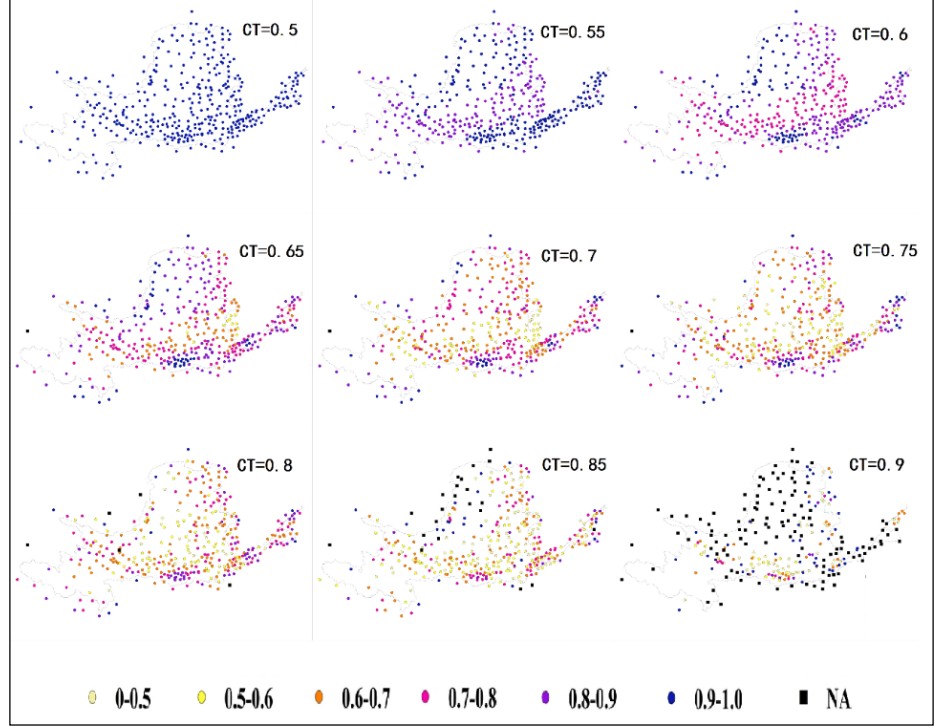

**Figure 2.** Clustering coefficient values of a P-network for nine threshold values.

As expected before, the clustering coefficient of a station in the network decreases with an increasing correlation threshold (from 0.5 to 0.9); that is to say, there is an inverse relationship between them. The clustering coefficients of most stations are greater than 0.7 in the case of small thresholds (as shown in Figure 2, the threshold values are 0.5, 0.55 and 0.6, respectively). This also means that we cannot separate the stations to study the spatial connection of the network. However, the clustering coefficients of most stations are less than 0.7 in the case of high thresholds (as shown in Figure 2, the threshold values are 0.85 and 0.9, respectively). In particular, at CT = 0.9, over 11% of the stations in the P-network are completely isolated, which means that the network becomes increasingly fragmented and less meaningful. It is not hard to understand the inverse relationship. In fact, the higher the value of the correlation threshold, the fewer stations are "filtered out". As a measure of local density, the local clustering coefficient also becomes smaller. Despite this, this simple result helps us to choose a suitable (strict or loose) standard and get a suitable network.

Table 1 presents the percentage of stations falling under different ranges of clustering coefficient values (P-network). As shown in Table 1, there is no consistency in the trend of the threshold value for the number of stations with the same range of clustering coefficient values. For example, there is a positive correlation between the number of stations and the correlation threshold in the case that the CC range is 0–0.5 or Na, while there is generally a negative correlation in other cases. Remarkably, for the clustering coefficient range 0.9–1, the number of stations increases abnormally in the case of CT greater than 0.85, which is caused by network fragmentation as noted earlier. From the perspective of network stations, the clustering coefficients of most stations decrease with the increase of the correlation threshold. Based on the above results, the correlation thresholds have a great impact on the clustering coefficient, which change edges between stations and their corresponding neighbors. This is obvious, and most previous similar applications of rainfall analysis with complex network methodologies have produced the same finding [26,28].

**Table 1.** Clustering coefficient values for different thresholds (P-network).

| CC Range | Percentage of Stations within Each Clustering Coefficient Range for Different CT (%) | | | | | | | | |
|---|---|---|---|---|---|---|---|---|---|
| | 0.5 | 0.55 | 0.6 | 0.65 | 0.7 | 0.75 | 0.8 | 0.85 | 0.9 |
| 0–0.5 | 0 | 0 | 0 | 0 | 2.37 | 5.54 | 7.65 | 18.21 | 36.41 |
| 0.5–0.6 | 0 | 0 | 0 | 3.17 | 13.72 | 18.73 | 17.68 | 25.07 | 7.39 |
| 0.6–0.7 | 0 | 0 | 0 | 19.00 | 31.13 | 31.93 | 31.66 | 22.96 | 8.71 |
| 0.7–0.8 | 0 | 0 | 32.19 | 35.36 | 32.19 | 22.16 | 24.80 | 15.57 | 1.58 |
| 0.8–0.9 | 0 | 44.85 | 45.91 | 30.08 | 13.72 | 14.78 | 13.72 | 6.07 | 3.17 |
| 0.9–1 | 100 | 55.15 | 21.90 | 12.14 | 6.60 | 6.60 | 3.17 | 6.33 | 11.61 |
| Na | 0 | 0 | 0 | 0.26 | 0.26 | 0.26 | 1.32 | 5.80 | 31.13 |

We discuss the network built with the Pearson correlation method above. We study the precipitation network constructed with the Spearman rank correlation method and compare similarities and differences between the two kinds of networks. Figure 3 and Table 2 present the clustering coefficient values of the S-network for nine threshold values and count the number of stations within each clustering coefficient range for different CT. It is obvious that the clustering coefficient values of the S-network are larger than that of the P network under the same threshold. This is because the Spearman correlation coefficient is based on the ranked values for each variable rather than the raw data, which increases the number of edges between stations and allows for more complex (yet monotonic) relationships. However, the relationship between threshold and number of stations shows no fundamental change, which is similar to the case of the P-network. Due to the increase in the number of edges, a higher threshold is applied to understand more details of the network properties. Compared with the P-network, the S-network has less information in the same threshold range. Therefore, we recommend the Pearson correlation coefficient method to build the network.

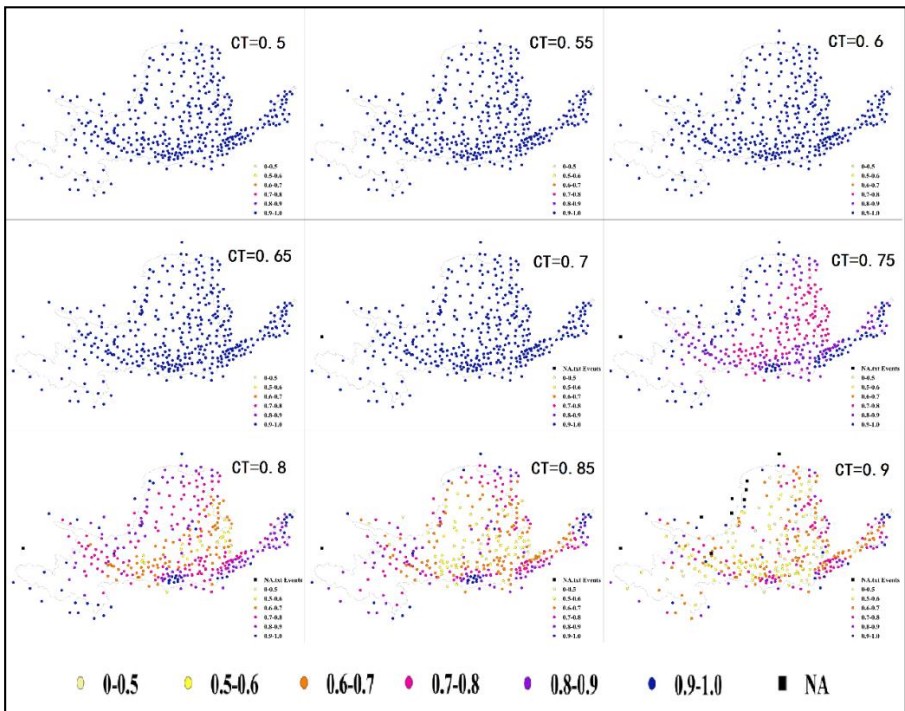

**Figure 3.** Clustering coefficient values of the S-network for nine threshold values.

**Table 2.** Clustering coefficient values for different thresholds (S-network).

| CC Range | Percentage of Stations within Each Clustering Coefficient Range for Different CT (%) | | | | | | | | |
|---|---|---|---|---|---|---|---|---|---|
| | 0.5 | 0.55 | 0.6 | 0.65 | 0.7 | 0.75 | 0.8 | 0.85 | 0.9 |
| 0–0.5 | 0 | 0 | 0 | 0 | 0 | 0 | 0 | 2.37 | 10.82 |
| 0.5–0.6 | 0 | 0 | 0 | 0 | 0 | 0 | 6.33 | 15.83 | 19.79 |
| 0.6–0.7 | 0 | 0 | 0 | 0 | 0 | 0 | 19.79 | 29.02 | 30.61 |
| 0.7–0.8 | 0 | 0 | 0 | 0 | 0 | 27.70 | 34.04 | 25.59 | 19.79 |
| 0.8–0.9 | 0 | 0 | 0 | 0 | 0 | 41.95 | 27.97 | 19.00 | 10.29 |
| 0.9–1 | 100 | 100 | 100 | 100 | 99.74 | 30.08 | 11.61 | 7.92 | 6.33 |
| Na | 0 | 0 | 0 | 0 | 0.26 | 0.26 | 0.26 | 0.26 | 2.37 |

It is worth noting that, no matter whether P-network or S-network, the spatial distribution of clustering coefficient values has no fundamental change. As shown in Figures 2 and 3, although the clustering coefficients of each point vary over different correlation thresholds, it seems that there is no fundamental impact on the spatial distribution of clustering coefficient values. For example, the clustering coefficients of stations in some regions are always higher than those in other regions, based on visual inspection, and it appears high–low–high from southeast to northwest. Results indicate that network properties change as a function of correlation threshold. However, something (e.g., network structure) has no fundamental change.

We attempt to interpret the relationships of clustering coefficients with station properties (e.g., latitude, longitude, and elevation) and precipitation properties, as suggested by Jha [26]. However, there is no obvious linear relationship between the local clustering coefficient and the selected station properties. However, we note that the spatial distribution of clustering coefficient values is similar to the annual average precipitation in the YRB. As mentioned earlier, the annual average precipitation in the YRB decreases from the southeast to the northwest; see Figure 1. Take the case of the P-network: each station is drawn on Figure 4 according to the clustering coefficient values and annual average precipitation. The X-axis represents the average annual precipitation and the Y-axis represents the clustering coefficient values. As shown in Figures 2 and 4, it is obvious that the P-network is divided into three parts by the 400 and 600 mm rainfall contours at low correlation thresholds (0.5–0.7), and the

clustering coefficient values of the two ends are greater than in the middle part. However, as the threshold increases, this phenomenon gradually disappears, and it is difficult to divide it into three obvious parts. It is consistent with the cluttered spatial distribution of the clustering coefficient values at high CT in Figure 2.

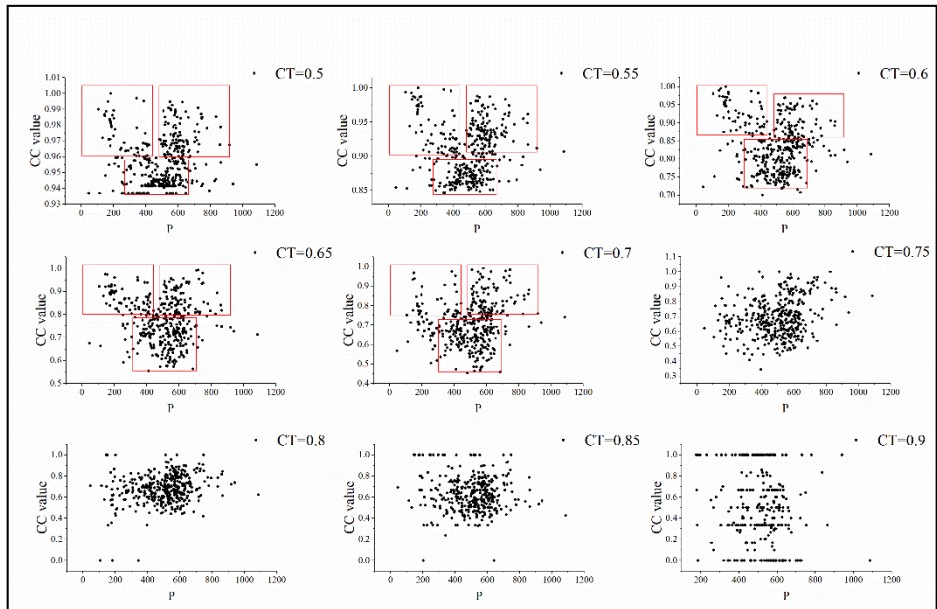

**Figure 4.** The clustering coefficient values versus annual average precipitation.

Figure 5 presents the spatial distribution of the normalized vertex degree of the P-network. Different from the distribution of clustering coefficient value, it appears low–high–low from southeast to northwest, which means that there is an inverse relationship between the spatial distribution of vertex degree and the clustering coefficient value in the P-network. As mentioned above, the clustering coefficient value is calculated by CC = 2E / k (k − 1). The greater the vertex degree of a station (i.e., the k in the formula), the greater the number of potential edges, that is, the denominator of the formula. If actual edges can be determined, the inverse relationship between the spatial distribution of vertex degree and the clustering coefficient value can be explained. Therefore, we need to figure out how the vertices link with each other. A useful index is the average degree of the neighbors of a given vertex.

We draw five plots of average neighbor degree versus vertex degree at CT = 0.5–0.7 in the P-network; see Figure 6. For CT = 0.5–0.6, it is obvious that there is a tendency for vertices of higher degrees to edge with vertices of lower degrees, and vertices of lower degrees tend to edge with vertices of higher degrees, which means that there is a negative relationship between average neighbor degree and vertex degree. Therefore, it is easy to explain why stations with small vertex degrees have high clustering coefficient values. For the case of CT = 0.65 and 0.7, different from the first three figures, the average neighbor degree has an obviously positive correlation with vertex degree. Nevertheless, almost all vertices of lower degrees are above the blue line (y = x), and most vertices of higher degrees are below the blue line, which still supports the conclusion that there is an inverse relationship between the spatial distribution of vertex degree and clustering coefficient value.

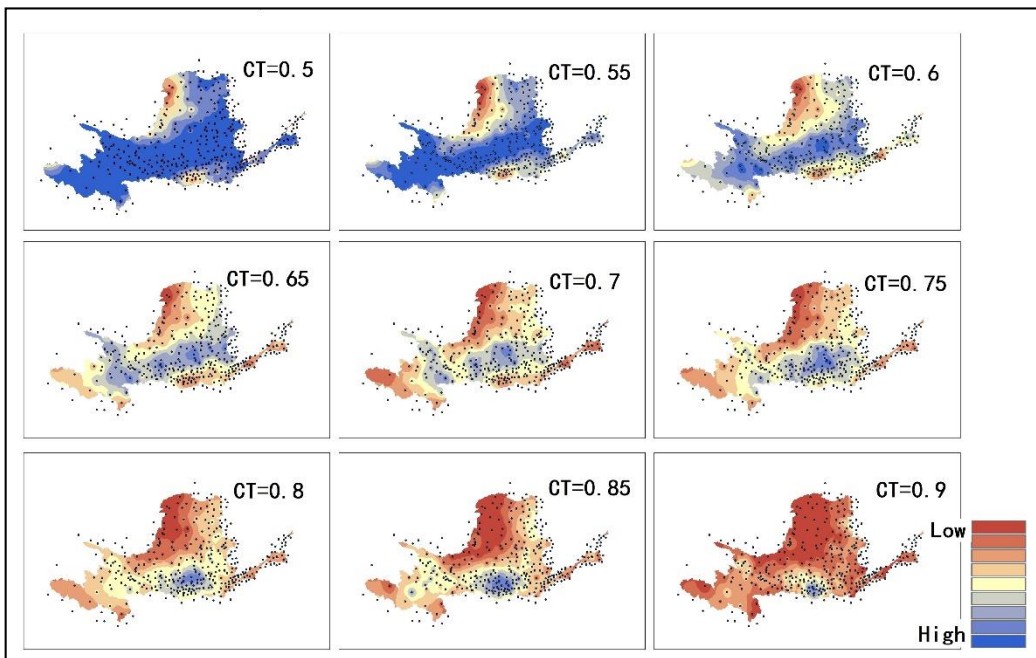

**Figure 5.** The spatial distribution of vertex degree (P-network).

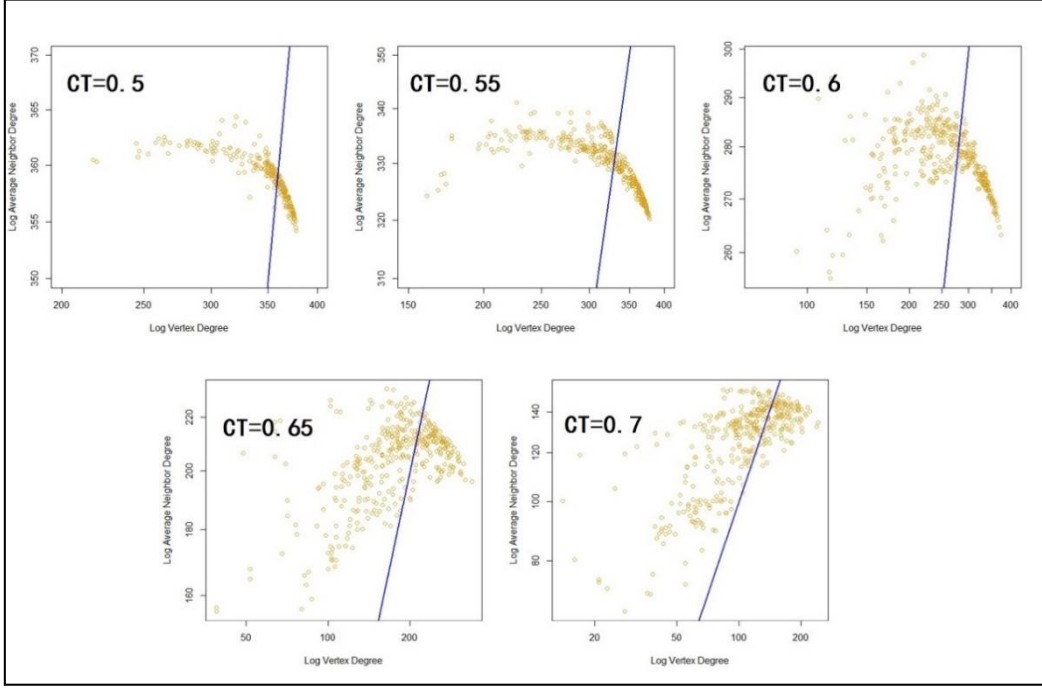

**Figure 6.** Average neighbor degree versus vertex degree (log–log scale).

Notably, selective linking among vertices, according to a certain characteristic or characteristics, is termed assortative mixing. Generally, this includes two aspects: vertices of lower (higher) degrees tend to edge with similar or different vertices; vertices of lower (higher) degrees have a higher (lower) clustering coefficient [41]. In fact, Newman [42,43] found that social networks differ from most other types of networks, including technological and biological networks, in two important ways. First, the vertices in the network that have many connections tend to be connected to other vertices with many connections, and second, they show negative correlations between vertex degree and clustering coefficient. It is interesting that the P-network has the two opposite characteristics of a social and nonsocial network at different thresholds. It is preliminarily considered that most stations' vertex

degrees are high, and the differences between them are not very large; it can be seen from Figure 6 that most points are concentrated near the blue line. The difference becomes gradually larger as the correlation threshold increases and the small-world property becomes obvious.

Additionally, we are also interested in which stations are more important for modeling catchment processes in future studies, which is not very relevant to the topic of this study. The P-network is divided roughly into three subnetworks by the rainfall contours of 400 mm and 600 mm; see Figure 7. Interior vertex degree (the number of edges with vertices in the same subnetwork) is useful for vertex importance evaluation to find important vertices in the subnetworks, which helps to optimize the precipitation network structure and interpolate the missing station data. We count the number of occurrences in the top 10% of interior vertex degrees to evaluate the importance of a station. Stations with over three occurrences are listed in Table 3 (vertices with a circle in Figure 7), which can be considered as important vertices in the subnetworks.

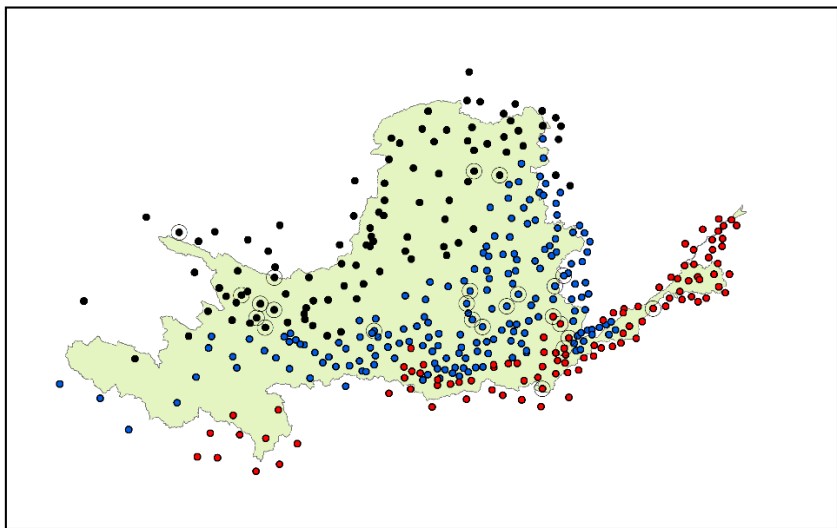

**Figure 7.** Partitioning of the P-network obtained from the clustering coefficient.

**Table 3.** Important vertices in subnetworks.

| Sub-Network 1 | | Sub-Network 2 | | Sub-Network 3 | |
|---|---|---|---|---|---|
| **Station** | **Frequency** | **Station** | **Frequency** | **Station** | **Frequency** |
| 52877 | 5 | 53848 | 4 | 53970 | 5 |
| 52787 | 4 | 53845 | 3 | 53975 | 5 |
| 52645 | 3 | 53859 | 3 | 57077 | 4 |
| 52866 | 3 | 53864 | 3 | 53978 | 3 |
| 52874 | 3 | 53872 | 3 | 54904 | 3 |
| 52876 | 3 | 53875 | 3 | 57071 | 3 |
| 52972 | 3 | 53910 | 3 | | |
| 53543 | 3 | 53942 | 3 | | |
| 53553 | 3 | 53946 | 3 | | |

*4.3. Network Architecture*

Work on the mathematics of networks has been a research hotspot in recent years, and focuses on finding statistical properties to characterize the structure and behavior of networked systems and creating models of networks. In recent years, a large number of empirical studies have been carried out on the topological features of many networks in the real world, and a variety of network mathematic models have been proposed, such as regular networks [44,45], random networks [46–48], small world networks and scale-free networks [49–52]. In fact, many networks in the real world, including hydrological networks and climate networks, meet the definition of a small-world network.

Tsonis [23] considered global climate as a network of many dynamical systems and found that the network has properties of small-world networks. Halverson [27] found that daily streamflow data in Canada displays properties consistent with small-world networks. In this section, we assess the significance of the small-world properties of precipitation networks in the YRB and try to prove that the choice of two correlation methods cannot lead to fundamental changes in the network architecture.

A typical approach for evaluating small-world behavior [34] is to compare the clustering coefficient and average shortest path length in an observed network to what might be observed in an appropriately calibrated classical random graph. Recalling the two properties of small world networks, we should expect under such a comparison—if indeed an observed network exhibits small-world behavior—that the observed clustering coefficient exceeds that of a random graph, while the average path length remains roughly the same.

Following the proof method above, we begin by computing the degree distribution of the precipitation networks and the expected degree distribution of random networks that have the same number of vertices and edges according to the Erdos–Renyi model [36]; see Figure 8.

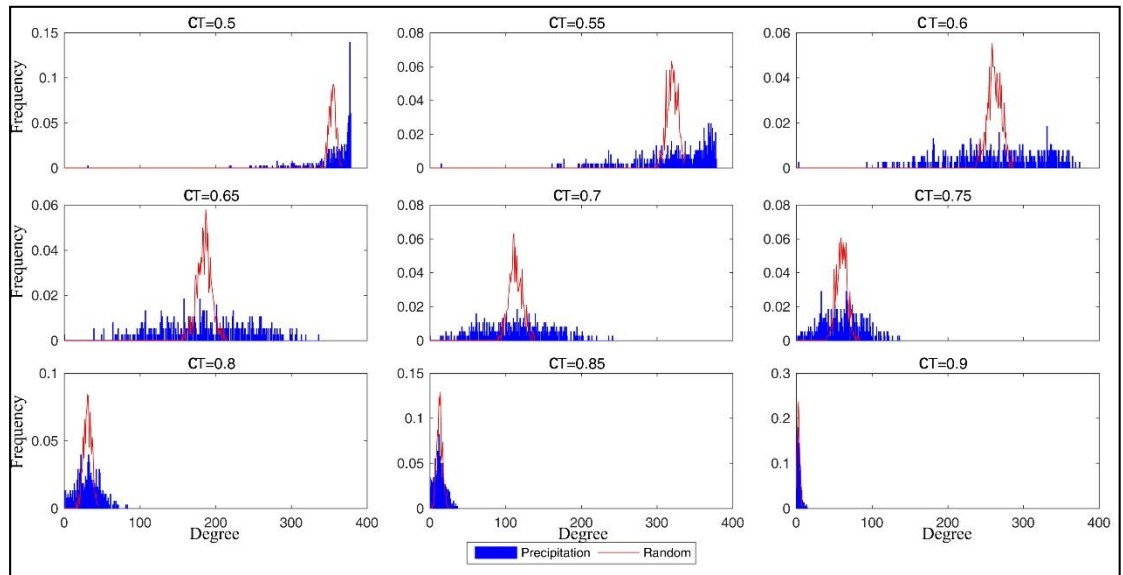

**Figure 8.** Degree distribution of the P-network and random networks.

Blue bars in the figures are discrete representations of the degree distribution of the precipitation networks, and red curves represent the expected degree distribution of random networks having the same edges and vertices. The degree distribution undergoes obvious changes when the CT is varied. For P-networks, the degree distribution of the precipitation network is asymmetrical in the case of small thresholds (CT = 0.5 / 0.55), which are characterized by a high right wing. As noted above, small thresholds make most stations' degree high and it is not surprising that the right wing is high. Meanwhile, in the case of higher thresholds (CT = 0.85 / 0.9), the degree distribution is also asymmetrical because of network fragmentation. Barring the two situations above, the degree distribution is approximately symmetrical with bound and noisy wings. As for the S-network, when CT ≤ 0.7 or CT = 0.9, the degree distribution is asymmetrical, which is consistent with the result above that the Spearman correlation method increases the number of edges between stations. From Figure 8, it is obvious that there is some resemblance between the degree distribution of the precipitation network and a random network. However, the random network has a narrow and high peak and a low tail in comparison.

We compute the global clustering coefficient and the average path length for different thresholds (see Table 4).

**Table 4.** The global clustering coefficient and the average path length for the P-network.

| CT | P-network | |
|---|---|---|
| | **Apath** | **CC** |
| 0.5 | 1.0628 | 0.9549 |
| 0.55 | 1.1525 | 0.8976 |
| 0.6 | 1.3079 | 0.8090 |
| 0.65 | 1.5178 | 0.7241 |
| 0.7 | 1.8325 | 0.6468 |
| 0.75 | 2.4349 | 0.6380 |
| 0.8 | 3.7052 | 0.6408 |
| 0.85 | 6.6712 | 0.5826 |
| 0.9 | 8.8586 | 0.4719 |

Then, we count the number of vertices and edges in the precipitation network for different thresholds and generate classical random networks of this same order and size. For each one, we compute its global clustering coefficient (CC) and average path length (apath). To eliminate some uncertain factors, the steps required to simulate draws of classical random networks need to be repeated 1000 times. The minimum, maximum and average values of apath and CC for each network are shown in Table 5.

**Table 5.** The average path length and the global clustering coefficient for a random network.

| CT | P-random Network Apath | | | P-random Network CC | | |
|---|---|---|---|---|---|---|
| | **Min** | **Mean** | **Max** | **Min** | **Mean** | **Max** |
| 0.5 | 1.0628 | 1.0628 | 1.0628 | 0.9371 | 0.9372 | 0.9373 |
| 0.55 | 1.1525 | 1.1525 | 1.1525 | 0.8473 | 0.8475 | 0.8477 |
| 0.6 | 1.3072 | 1.3072 | 1.3072 | 0.6924 | 0.6928 | 0.6932 |
| 0.65 | 1.5099 | 1.5099 | 1.5099 | 0.4891 | 0.49 | 0.491 |
| 0.7 | 1.7006 | 1.7006 | 1.7006 | 0.298 | 0.2994 | 0.3008 |
| 0.75 | 1.8406 | 1.8407 | 1.8408 | 0.1568 | 0.1594 | 0.1616 |
| 0.8 | 1.9849 | 1.9882 | 1.9923 | 0.0785 | 0.0823 | 0.0862 |
| 0.85 | 2.5963 | 2.6023 | 2.6087 | 0.0284 | 0.0341 | 0.0401 |
| 0.9 | 5.0019 | 5.2504 | 5.5338 | 0 | 0.008 | 0.0204 |

In order to facilitate the analysis, Figure 9 is drawn based on the simulation results in Tables 4 and 5, where apath_random means the average path length of the simulated random network; CC_random means the global clustering coefficient of the simulated random network. With regard to the P-network, the clustering coefficient decreases from 0.81 to 0.64 as CT increases from 0.6 to 0.8, which is always much larger than the average clustering coefficient value of the simulated random network. Meanwhile, for values of CT < 0.8, the average path length remains only slightly higher than what would be expected for the simulated random network. It is completely clear that the P-network has small-world properties for any value of CT between 0.6 and 0.8, which satisfies the criterion that CC >> CC_random and apath ≥ apath_random. For values of CT < 0.6, the threshold is too small to outstand each station's feature. The overwhelmed and distorted network is similar to a random network, where stations have the same edges. For values of CT > 0.8, the fragmented and meaningless network excludes many important edges. In summary, we suggest that the precipitation network in the YRB has the architecture of a small-world network in the appropriate threshold range. Moreover, the perturbations of the correlation methods used for edge definition do not bring a fundamental change in network topology.

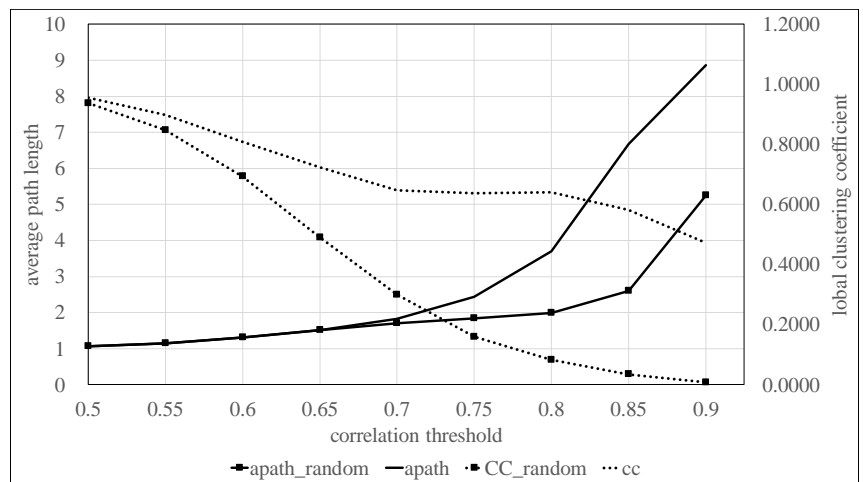

**Figure 9.** Comparison between the precipitation network and a random network.

## 5. Conclusions

The present study applies the concept of the complex network to examine the spatial connections and network architecture of a precipitation network in the YRB. Sensitivity tests, including correlation thresholds and correlation methods, provide some interesting results about the precipitation network. The results indicate that choice of the correlation threshold significantly influences the local clustering coefficient in an inverse correlative relation. The clustering coefficient of a station in the network decreases with the increasing correlation threshold. The network becomes increasingly fragmented and less meaningful at a high correlation threshold. The choice of the correlation method has no obvious influence on the precipitation network, but the fact that the S-network allows more edges between stations convinces us that the Pearson correlation coefficient method is more suitable for network construction. We also find that the spatial distribution of the clustering coefficient appears high–low–high from southeast to northwest; however, by contrast, the spatial distribution of the vertex degree appears low–high–low. In addition, there is an inverse relationship between average neighbor degree and vertex degree at low CT, and a positive correlation at high CT. Some important vertices are also found by interior vertex degree in the three sub-networks. Furthermore, we studied the precipitation network architecture and found the small-world properties in the appropriate threshold range (CT $\in$ (0.6, 0.8)).

Although this study is still preliminary, it offers another perspective to study the connections between stations. Since the correlations between stations are independent of geographical distance, it is important and necessary to combine the concepts of complex networks with traditional interpolation and extrapolation methods to develop a more reliable model. Therefore, we attempt to interpret the relationships of clustering coefficients with physical properties, but the result is not satisfactory, which is also an urgent problem to be solved. We hope to report the details of these studies in the near future.

**Author Contributions:** Conceptualization, F.L., K.Z. and Y.X.; Data curation, Y.D. and X.S.; methodology, Y.X. and F.L.; software, X.S. and Y.X.; formal analysis, Y.X., F.L. and X.S.; writing—original draft, Y.X.; writing—review and editing, Y.X., F.L., K.Z. and Y.D.; All authors have read and agreed to the published version of the manuscript.

**Funding:** The research is financially supported by the National Key Research and Development Plan of China (No. 2017YFC0404401) and the National Natural Science Foundation of China (Grant No.: 51679252 and 51409246).

**Conflicts of Interest:** The authors declare no conflict of interest.

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
