# Peer review of "Exploring the Clustering Property and Network Structure of a Large-Scale Basin’s Precipitation Network: A Complex Network Approach"

_water, doi:10.3390/w12061739_

Round 1

Reviewer 1 Report

This work “Exploring the clustering property and network structure of large-scale basin’s precipitation network: A complex network approach”. In this manuscript, authors carried out a sensitivity test including correlation thresholds and correlation methods and found that the choice of methods has little effect on the precipitation networks. However, correlation thresholds significantly affected vertex degree and clustering coefficient values of precipitation networks. Although, the work is interesting, some points should be improved: • The abstract of the article can be more concise and must be restructured. Name few of the applications i.e., where this sensitivity test can be used for the better understanding of readers at the end of abstract. • There are few grammatical errors throughout the paper, which need to be corrected. Try to avoid unnecessary long sentences. • Literature review is ambiguous; include some more recent state of the art papers in Literature review for better understanding. • The Introduction part of the article must be revised to make it better structured for the readers. Try to explain the previous work related to different aspects of the current research and connect it with the problem statement in the end i.e. identifying the gap and why was this model necessary to develop. An intense revision is required in this section. • In the last paragraph of the introduction section, mention the novelty of this paper with previous state of the art research. Rather than mentioning the results/conclusions of the manuscript. Moreover, mention the applications of this work. • If possible, compare the numerical data with the data already present in literature for validation in graphical form. • Fig.1 is very simple, it can be avoided. • Mention the source of Fig.2 in the caption of the figure. • Sections 2.1-2.4 are adding something to the scientific community? Or are they too simple, like basics? Explain this point that what are they contributing? • Explain this critical point in detail that choice of the correlation thresholds influences significantly the local clustering coefficient in an inverse correlative relation and what is the physics behind this?

Reviewer 2 Report

This a well motivated study which introduces ideas of complex networks for application in precipitation modelling. It offers a new and interesting perspective with the present work and prospect fro future studies, while attaining some good handle on the suitability of this approach via establishing the Pearson R-coefficient method as more appropriate means to construct networks and ultimately improve statistical precipitation modelling. The paper will benefit from a thorough review for some typos minor grammatical and language usage. For example, in line 127, it should be k=4, instead of k4=4, and there may be instances of such typos in the paper. Upon these review and minor corrections, I recommend the publication and wish the authors success in their upcoming studies.

Round 2

Reviewer 1 Report

The paper in good shape now and can be published in the current form.